# Ecology of Nontuberculous Mycobacteria

**DOI:** 10.3390/microorganisms9112262

**Published:** 2021-10-30

**Authors:** Joseph O. Falkinham

**Affiliations:** Department of Biological Sciences, Virginia Tech, Blacksburg, VA 24061, USA; jofiii@vt.edu; Tel.: +1-540-231-5931

**Keywords:** nontuberculous mycobacteria, soils, natural and human-engineered water systems, estuarine, hydrophobic, aerosolization, surface microlayer, biofilm formation

## Abstract

Nontuberculous mycobacteria (NTM) are opportunistic human pathogens that are widespread in the human environment. In fact, NTM surround humans. The basis for their widespread presence in soils and natural and human-engineered waters lies primarily in their disinfectant resistance, biofilm formation, and adaptability to fluctuating environmental conditions. As NTM in drinking water surround humans, a major route of infection is through aerosols. The characteristics of NTM, including resistance to disinfection, adherence to surfaces and biofilm formation, present challenges to contemporary water treatment processes developed for control of *Escherichia coli* and fecal coliforms.

## 1. Introduction to Nontuberculous Mycobacteria (NTM)

Nontuberculous mycobacteria (NTM) comprise a large collection of *Mycobacterium* species that are human, bird, fish, and animal pathogens and whose source of infection is the natural and human-engineered environment. As such, it is likely that humans, animals, fish, and birds are regularly exposed to NTM. Further, that observation leads to the hypothesis that NTM are quite adaptable and hardy. This brief review provides some evidence in support of that hypothesis.

For humans, NTM are opportunistic pathogens, infecting individuals with pre-existing conditions that make them more susceptible [1]. Behavioral risk factors include smoking, excess alcohol consumption, and exposure to dusty occupations (e.g., farming and mining). Genetic factors include cystic fibrosis, α-1-antitrypsin deficiency, and deficiencies in the production of immune-signaling proteins. Infection by HIV leading to AIDS is also a risk factor for NTM infection (bacteremia). At present, the largest group of NTM-infected individuals, estimated at 181,000 in the United States [2], are older, slender, taller women whose primary risk factor has not been identified. Transmission of NTM from the environment to humans can occur via generation of aerosols from NTM-containing waters and inhalation of NTM-laden aerosols; for example, in showers [3]. Dust inhalation, from NTM-colonized soil or potting soil mix, is also a means of transmission and infection [4]. Skin NTM infection can be the result of exposure to NTM-containing water, soil, and infected fish or animals [5]. Drinking liquids or foods that can occur as a result of gastroesophageal reflux disease (GERD) may also be a route of NTM transmission and infection [6].

## 2. Habitats Occupied by NTM

### 2.1. Soils

Soils are rich sources of NTM, with numbers as high as 1 million cells per gram [7,8]. NTM were recovered from alpine and subalpine soils of the Jura Mountains [9]. In particular, peat-rich soils such as sphagnum bogs of the northern latitudes [10,11] and acidic, brown-water swamps of the eastern and southeastern coasts of the United States [12] harbor high numbers of NTM. It is likely that water run-off from soils leads to the introduction of soil-originating NTM in surface waters that, in turn, enter water treatment plants, distribution systems, and hospitals, long-term care facilities, office buildings, apartments, condominiums, and single-family houses. High cell-surface hydrophobicity of NTM drives the adherence of cells to soil particulates as elevated water turbidity is positively associated with NTM presence [13]. Further, the reduction in turbidity in water treatment plants as a consequence of settling leads to a 50% reduction in water-borne NTM [14].

### 2.2. Sediments

Although there have not been any systematic studies, NTM have been recovered from sediments [7]. As NTM cells are hydrophobic, they adhere to soil particles that can associate with ever larger particulates until they settle to the bottom of ponds. NTM survival in sediments suggests that they can survive under low oxygen levels, or even anaerobiosis. The growth of *Mycobacterium avium*, *Mycobacterium intracellulare*, and *Mycobacterium abscessus* strains at low oxygen levels [15] is consistent with the observation that NTM survive and persist in sediments. Further, unlike *Mycobacterium tuberculosis*, an obligate human pathogen, strains of each of those three species could survive rapid shifts to anaerobiosis [15].

### 2.3. Dusts

NTM adhering to soil particles can be transmitted as dusts. Indoor dust samples have been shown to harbor NTM flora [8,16]. Biofilms collected from archaeological sites in Mexico have been shown to harbor NTM [17]. The presence of NTM in dust and in archaeological sites supports the notion that NTM are resistant to desiccation (see below). Peat-rich, potting soil which can be purchased in do-it-yourself stores in North America carries upwards of 1 million NTM cells per gram [4]. To overcome the challenge of isolating NTM from soil samples by culture where NTM colonies are often overgrown by faster-growing microorganisms (almost all), 100 g samples of potting soil from NTM-infected patients was collected and dropped 1 m inside a closed chamber. The resulting dust particles were captured using an Andersen 6-Stage Cascade sampler [18] and shown to carry NTM [4]. The fact that the isolated NTM were identified as belonging to the same *Mycobacterium* species as the isolates from the patients providing the potting soil sampled, suggests that dusts generated from handling the potting soils were the source of their NTM pulmonary infections [4].

### 2.4. Estuaries

Estuaries, namely bodies of coastal water subjected to tidal inflows and outflows of ocean and fresh water are also habitats that harbor high numbers of NTM. Specifically, 1000–10,000 NTM colony-forming units/mL were recovered from water samples collected in the Chesapeake and Delaware Bays [19]. Their presence in estuaries suggests that NTM, or some NTM species, can tolerate fluctuations in NaCl concentrations. NTM are not recovered from ocean water samples [19] and fail to grow in ocean water (>3% NaCl), but can grow in brackish water (1–2% NaCl) [20].

### 2.5. Surface Waters

Surface waters across the United States and Canada yield NTM [14,19] and as surface waters are used as sources for drinking water, NTM are found in treatment plants, distribution systems, and premise plumbing [14]. Pond, lake, stream and river water samples have all yielded NTM suggesting that NTM can survive, persist, and even grow in those habitats. My lab’s original sampling was initiated based on evidence of high frequencies of skin sensitivity to purified protein derivatives (i.e., PPD-B) were found in the southeastern United States [21,22]. As we pursued those studies, my colleague Bruce Parker, introduced me to the presence of surface microlayers in natural waters. The microlayer comprises the surface 10 μm and can be collected by carefully laying filter paper on the water surface and extracting the adherent inorganic, organic and microbial constituents [23] or by submerging a glass plate below the surface and slowly drawing the plate upwards [24]. In addition, large volumes of the surface microlayer can be collected by instrumentation [25]. The chemical and microbiological composition of surface microlayers in water differs considerably from that of the bulk water [23,25]. Concentrated in the surface microlayer are organic compounds (e.g., oils) [23,25] and hydrophobic microorganisms [24,25], especially NTM.

Although high numbers of *Mycobacterium scrofulaceum* were found, along with lower numbers of other NTM [19], later surveys of the southeastern United States failed to document the presence of *M. scrofulaceum* [7]. In fact, *M. scrofulaceum* has disappeared. Surveys of childhood cervical lymphadenitis showed that *M. scrofulaceum* was the predominant NTM isolated, but after 1985, it was not isolated and, instead, was replaced entirely by *Mycobacterium avium* [26]. *M. scrofulaceum* cannot be missed as it forms large 2 mm diameter, domed bright yellow colonies that form well before those of its relatives, *Mycobacterium avium* and *Mycobacterium intracellulare*. It was only in 2000 that we had data that suggested a possible explanation. In 1987, a paper in Science documented the consequences of the “Clean Water Acts”, enacted into law by the U.S. Congress starting in 1970 [27]. The “Clean Water Acts” provided funds for the improvement of water treatment plants with the express purpose of reducing microbial contamination of rivers and lakes and, further, provided cleaner water to the citizens of the United States [27]. One result of that legislation was widespread implementation of disinfection of surface water for drinking, commonly by chlorination [26]. Measurements of chlorine and chloramination susceptibility of *Mycobacterium avium*, *Mycobacterium intracellulare*, and *Mycobacterium scrofulaceum* showed that *M. scrofulaceum* was considerably more chlorine susceptible compared to either *M. avium* or *M. intracellulare* [28]. I suggest that the implementation of widespread disinfection led to the replacement of *M. scrofulaceum* by *M. avium*.

As NTM are relatively slow growing compared to other bacteria, there must be adaptations to prevent disappearance from flowing habitats. One such adaptation is surface adherence and biofilm formation [29]. NTM are slow growing due, in part, to the diversion of energy (i.e., ATP) from cell multiplication to the synthesis of the long-chain (C_60_–C_80_) lipid-rich outer membrane. The presence of that almost wax-like surface, NTM are concentrated in the surface layers of bodies of water [24,25]. Air bubbles, rising through the water column, collect the water-hating NTM cells, and concentrate those cells in the surface microlayer, where organic molecules also collect.

### 2.6. Ground-, Well- and Spring-Water

Three collections of data suggest that one source of NTM is not ground-, well-, or spring-water. First, in a survey of waters collected from wells in Montgomery County in southwestern, Appalachian, Virginia, we failed to find NTM in a large collection of samples [30]. That result was not due to the absence of NTM in all waters as we regularly isolate NTM from drinking water samples in the same county; even the University which is an excellent source of NTM-colonized water samples. The Montgomery County, Virginia data were not unusual, for in a survey of drinking water and biofilm samples collected from NTM-infected patient home plumbing, homes whose water source was wells, seldom yielded NTM (or in low numbers). In contrast, NTM-infected patient homes whose water source was piped systems yielded a wide variety of NTM in numbers above 100–1000 colony-forming units per area mL [31]. Finally, in response to NTM-infected patient’s questions, about what type of water to drink if one cannot filter, irradiate, or boil water, I went to the local grocery stores and bought containers of “distilled”, “purified”, and “spring” water. Both “distilled” and “purified” (whatever that is) yielded substantial numbers (10–1000 CFU/mL) and species variety of NTM. “Spring” water samples yielded 10 CFU/mL or less. That is why I recommend that individuals infected with NTM or concerned that they have a risk factor for NTM disease drink bottled “spring” water while travelling away from home. There are alternatives, namely water bottles that filter or UV-irradiate the water in the container; the water coming from the bottles is free of NTM [32].

## 3. Transmission of NTM from Environmental Habitats

### 3.1. Aerosolization

NTM, by virtue of their hydrophobic surface, are readily aerosolized from waters [33]. *M. avium* pulmonary infection has been traced to the presence of *M. avium* in a showerhead [3], to NTM in a humidifier [34,35], and *M. avium* in hot tubs [36]. In addition, *Mycobacterium chimaera*, a member of the *M. avium* complex (MAC), bacteremia in cardiac surgery patients throughout the world was traced to its presence in an operating room device called a heater-cooler [37]. NTM cells, as exemplified by *M. avium*, are collected by air bubbles rising in a water column and upon reaching the surface the bubbles burst resulting in the ejection of water droplets above the water [33,38]. Cells of NTM and other waterborne pathogens, such as *Legionella pneumophila* and *Pseudomonas aeruginosa*, can be collected above (1 cm) water surfaces. Further, the density of NTM cells in the ejected droplets is orders of magnitude higher than the density in the bulk suspension [33]. A proportion of the droplets are of a size able to be transferred via air currents throughout an operating room [37]. NTM were collected by placement of inverted Petri dishes containing mycobacterial agar medium 10 cm above the surface of the James River in Richmond, Virginia [24] documenting the water-to-air transfer of NTM.

### 3.2. Dust Generation

The presence of NTM in dusts collected from homes [8,16,39] strongly suggest that inhalation of dusts inside homes can be sources of NTM infection. Further, the fact that some NTM species are desiccation tolerant (see below) reinforces that hypothesis.

### 3.3. Swallowing

The widespread presence of NTM in drinking water suggests that drinking water may be a source of NTM infection; specifically, in individuals with esophageal gastric reflux disease (GERD). A study of a matched pair of patients showed that the prevalence of NTM pulmonary infection was higher in patients with GERD compared to matched patients without GERD [6]. Thus, swallowing liquids with NTM is a risk factor for NTM disease.

### 3.4. Surface Contact

NTM infections are not limited to pulmonary disease, but skin infections have been reported. The classic example of NTM skin infections are granulomas caused by *Mycobacterium marinum* in individuals who actively maintain fish tanks [5] or commercial fishermen [40]. Infection is associated with exposure of abrasions or cuts in the hands to the fish or water [5,40]. A variety of fish species, included ornamental and farmed, food fish, can be infected by NTM. The infected fish usually display granulomas on the gills. Not only *M. marinum*, but other *Mycobacterium* species have also been reported as infecting fish; usually found in fish farms or public aquaria [40].

## 4. NTM Characteristics Contributing to Environmental Survival

### 4.1. Hydrophobicity

The major contributor to NTM survival and persistence in the environment is cell surface hydrophobicity. Cell surface hydrophobicity of NTM is due to the presence of the lipid-rich outer membrane [41]. It is responsible, in part, to the concentration of NTM cells in the surface microlayer [24,25], the concentration of NTM in water droplets ejected from surface waters [33], adherence to particles in soils [14], adherence to natural (e.g., rocks) and human-engineered (e.g., pipes) surfaces to prevent washout, and disinfectant resistance [28,42]. However, the hydrophobic outer membrane has a cost. NTM cell growth rate is reduced by the diversion of resources to long-chain lipid synthesis (C_60_–C_80_) and to the reduced rate of transport of nutrients into cells [41].

### 4.2. Humic and Fulvic Acid Growth Stimulation

In a study of NTM presence in the Dismal Swamp of coastal Virginia, it was discovered that NTM numbers were correlated with organic matter content [12]. Further investigation demonstrated that the correlation was increased when NTM numbers were measured against humic and fulvic concentrations [43]. Following that study a laboratory growth study was conducted to determine the basis for the correlation. Growth of *M. avium*, *M. intracellulare*, and *M. scrofulaceum* strains was significantly increased in the presence of humic and fulvic acids [43]; yet there was no consumption of humic or fulvic acids. We continue to investigate the basis for the humic and fulvic acid-growth stimulation, perhaps as a chelator of metals, especially zinc. In spite of our current failure to identify the basis for the growth stimulation, this activity certainly supports the hypothesis that NTM are adapted to growth in coastal swamps that are rich in humic and fulvic acids.

### 4.3. Salt Tolerance

A high percentage of single-county, male residents of the southeastern United States have skin test reactions to the mycobacterial antigens PPD-B [21] and PPD-G [22]. The fact that the percentage of reactors was highest at the coast and fell as distance from the coast inland increased, suggested that the ocean was the source of NTM. However, that proved not to be the case. First, ocean water samples collected on the eastern coast of the United States did not yield any NTM [19]. Second, prompted by that result and the fact that water samples collected from the Chesapeake and Delaware Bays had high numbers of NTM, we measured the growth of NTM strains in natural fresh water samples, brackish water samples collected from the Chesapeake and Delaware Bays, and ocean water samples (i.e., >3.0% NaCl). The results documented NTM growth in fresh and brackish water samples and a failure of ocean water to support NTM growth [20]. The fact that NTM were recovered from brackish estuarine water samples with salinities between 0.1 and 2.5% suggests that NTM are estuarine bacteria [19,20]. Further, NTM have no novel nutrient requirements and can grow in natural water samples of low organic concentrations [20,43]; they are oligotrophs.

### 4.4. Desiccation Tolerance

Following the outbreak of *M. chimaera* bacteremia linked to a single manufacturer’s heater-cooler among patients who had undergone cardiac surgery [37], there was evidence from whole-genome sequencing that all the isolates were related [44]. Evidence that water from the Munich, Germany manufacturing facility contained the same strain of *M. chimaera* [45] suggested that the newly made instruments were tested for function before shipping. That testing would have required filling the instruments with water; namely local Munich water. Realization that the instruments were filled with *M. chimaera*-containing water, but then tested and drained before shipping throughout the world, brought another question forward. How did the colonizing *M. chimaera* cells remain viable during the long times required for shipping; for example, 3 weeks was recorded for shipping from Munich to York, Pennsylvania. We hypothesized that the *M. chimaera* cells rapidly adhered to the walls of the water circuit in the heater-coolers and formed biofilms. Evidence from measurements of *M. avium* and *M. intracellulare* relatives of *M. chimaera* showed that substantial biofilms were formed on stainless steel coupons within 1 h [29], a period of time likely to be similar to that required for functional testing. Accordingly, we measured the desiccation susceptibility of *M. avium*, *M. intracellulare*, *M. chimaera*, *M. abscessus*, and *M. chelonae* strains adhering to stainless steel coupons. Survival of the *M. avium*, *M. intracellulare*, and *M. chimaera* cells was remarkably high, staggeringly high for one: namely, 18% for an *M. avium* strain, 119% for an *M. intracellulare* strain, and 15–28% for 4 different isolates of *M. chimaera*, including two isolated from colonized heater-coolers. Survival of the *M. abscessus* (0.6%) and *M. chelonae* (<0.02%) was less [Falkinham, in preparation}. I conclude from the data that long-term survival is a characteristic of members of the *M. avium* complex, but not all NTM, and that is consistent with their survival, growth, and persistence in fluctuating natural and human-engineered environments.

## 5. Habitat Adaptation by NTM

### 5.1. Introduction

Adaptation, as contrasted to mutation, is a nonpermanent change in the characteristics of an organism. Classically amongst bacteria, adaptation is exemplified by a switch between phenotypes, such as surface antigens or serotypes. In some instances, it is possible to show that growth conditions influence the presence or absence of the adaptation.

I suggest that adaptations may be of more importance to NTM, than they are to other microorganisms. First, consider that NTM must survive, persist, and grow under continuously fluctuating conditions; the natural and human-engineered environments. Second, as NTM grow relatively slowly compared to other microbes, the spread of beneficial mutants would be limited to the point of extinction in a fluctuating environment. Third, although NTM grow relatively slowly, that does not indicate they have a slow metabolism. Quite the contrary, NTM metabolism is as rapid as that of *Escherichia coli.* NTM grow slowly, in part, because a great deal of energy is diverted to the synthesis of the long-chain lipid-rich outer membrane that comprise 30% of the cellular weight. The following are examples of NTM phenotypic changes to provide evidence of adaptation.

### 5.2. Colony Type Variation in M. avium

One well-established adaptation is the switch between rough, transparent and smooth colony types in *M. avium*. The rough, transparent colonies are small (1 mm diameter), irregular and transparent and hat-shaped and are almost invisible on agar medium. The smooth colonies are larger (2 mm diameter) and easily seen. The rough, transparent colony types are more virulent than their isogenic smooth variants. The growth rate of isogenic smooth colony-type cells is much faster than that of their transparent-colony-type segregants. The transparent variants are more hydrophobic, more virulent, and more antibiotic-resistant that its isogenic smooth variants [46]. In a population of transparent colonies, typically isolated from an infected patient, 1 of 1000 colonies will be the alternative smooth type. In like fashion, in a population of smooth colonies, one of a thousand will be transparent. The reversibility of the colony switch can be easily demonstrated by isolating a single smooth colony from a lawn of the transparent variant (easily seen) and growing the resulting pure smooth colony. That smooth colony isolate upon growth will yield transparent colonies at a frequency of 1/1000 (very difficult to see). As the smooth variants grow considerably faster than the transparent variants, long-term transfer in laboratories can lead to stock cultures all switching to smooth.

### 5.3. Temperature Tolerance in M. avium

In studies of NTM in household plumbing of NTM-infected patients, it was discovered that in homes whose water heater set temperature was 125 °F (50 °C) or less yielded NTM [31]. In contrast, if the water heater set temperature was 130 °F (55 °C) or greater, NTM were rarely recovered [31]. On that basis, I have advised NTM patients or individuals at risk for NTM infection (e.g., cystic fibrosis patients, immunocompromised, or taller, slender, older women) to raise their water heater set temperature to 130 °F (55 °C). Preliminary evidence from implementing that action in 10 of 40 *M. avium*-infected homes in a suburb of Philadelphia, PA, showed that *M. avium* disappeared from water samples collected in their homes by 8 to 12 weeks [47]. We continue to collect and analyze water samples from those homes as we are concerned that the imposition of a single selective action, namely high temperature, could result in emergence of temperature-resistant mutants of *M. avium*. Fortunately, none have appeared over the on-going, 3 year period of the trial.

However, measurement of the susceptibility of the household *M. avium* isolates to exposure to 65 °C showed that cells grown at 25 °C were entirely killed in 1 h at 65 °C. Cells from cultures grown at 42 °C, the highest optimal growth temperature for *M. avium*, survived exposure to 65 °C for greater than 2 h [48]. Growth at intermediate temperatures, 30 °C and 37 °C, also increased survival above that for cells grown at 25 °C [48]. That growth-temperature adaptation to high temperature survival is likely mediated by the increased production of membrane-associated trehalose [48], in agreement with studies of thermal tolerance in other bacteria.

## 6. Geographic Distribution of NTM in the United States

### 6.1. Distribution of NTM Inferred from Skin-Sensitivity Reactions

My laboratory’s starting point for our studies of NTM habitats and ecology was based on maps of the geographic distribution of PPD-B [21] and PPD-G [22] skin reactors in the United States. The PPD-G antigen was prepared from the Gause strain of *Mycobacterium scrofulaceum* and skin tests showed that it had the highest reactivity among southeastern U.S. residents [22]. Both maps showed that greater than 60% of southeastern, single county-male residents of the United States had positive skin tests. That data led to Dr. Howard Gruft, Director of the Tuberculosis Research Laboratory of the State of New York to recruit my colleague, Bruce Parker and I to collect water samples in the eastern United States and isolate, identify, and enumerate NTM. The results showed that NTM were in mostly all water samples collected in both the northeastern and southeastern United States, but not in ocean water [19,20]. As I now realize from looking back at our results and the frozen isolates, the majority of our isolates in the southeastern United States were *M. scrofulaceum*, the relatively rapidly growing, large, yellow-pigmented colony formers, that appear to have disappeared from the environment; likely due to chlorine-sensitivity as noted above [28]. The hypothesis that NTM were localized in the southeastern United States was not only disproven by our culture-based study, but by a call from a Cincinnati-based physician reacting to our 1980 publication. He asked how I could explain that AIDS patients across the United States were infected with *M. avium*, if NTM were localized in the southeastern United States. At the time, I simply had no answer. I now understand that the major source of NTM and *M. avium* is domestic, piped water systems in our homes, hospitals, and businesses.

### 6.2. Distribution of NTM Based on Isolation, Enumeration, and Identification of NTM

Briefly, NTM are widely distributed across the United States and in substantial culturable numbers [14,31]. Following the linkage of *M. avium* in a showerhead to pulmonary disease by DNA fingerprinting [3], a survey of showerheads across the United States was undertaken. That survey employing DNA isolation, rather than culture, showed that NTM were present in 70% of showerheads sampled and of those 70%, 30% had *M. avium* [49]. Based on that data, I think that surveys for NTM presence are unnecessary. Simply, NTM surround people.

### 6.3. NTM Distribution Influenced by Geology—The “Fall Line” of the Eastern, Coastal United States

There are scattered and unsystematic reports of unique geographic distributions of NTM species. In many instances, those geographic distributions are employed for naming *Mycobacterium* species, such as *Mycobacterium malmoense* whose case reports are centered in Sweden. Another example involves *Mycobacterium simiae*. It has been suggested that the Ogalala Aquafer that is a major water source for the lower central plains of the United States is the origin of *M. simiae*.

In Virginia, biologists and geologists are aware of the “Fall Line”. This is the break between the granite-based Appalachian Mountain region, the “Piedmont” and the sand-based coastal plain, the “Tidewater”. In addition to geologic differences, there are major differences in the flora of the two regions. The term “Fall Line” is based on the presence of rocks and rapids in the rivers flowing from the Appalachian Mountains to the Atlantic Ocean. The “Fall Line” was also the end of river navigation in colonial America as the boats could not travel above the falls. It was also where colonial cities appeared, for example Georgetown in the District of Columbia, Richmond in Virginia, Roanoke Rapids in North Carolina, and Savannah in Georgia. Florida is all coastal plain and New England is mostly Piedmont. For the most part, if one follows the route of Interstates 81 and 58 from Pennsylvania south to Alabama you are on the “Fall Line”.

Based on scattered data on the locations of NTM species infecting patients (e.g., homes and hospitals) or recovery of NTM species from the environment, *M. avium* is located in the Piedmont, west of the “Fall Line” and *M. abscessus* in the Tidewater, east of the “Fall Line”. I want to test that hypothesis thoroughly, as it might permit more rapid treatment decisions for NTM-infected patients. Specifically, *M. abscessus* presents a greater treatment challenge compared to members of the *M. avium* complex. Accordingly, I am in the process of collecting data and will be sampling water and soil samples in Virginia, bracketing the “Fall Line”.

## 7. Ecology of NTM in Household and Hospital Plumbing

### Indoor Plumbing—An Ideal Habitat for NTM

In this section, I wish to highlight the characteristics of NTM and to explain how those lead to the fact that plumbing is an ideal habitat for NTM. All contribute to the fact that humans are surrounded by NTM; especially as we spend a great deal of time indoors.

NTM enter structures through water pipes from municipal piped systems whose major source is surface water [14]. NTM are in surface waters due to drainage of soil particulates from soils and they grow as oligotrophs on low concentrations of organic matter. Although approximately 50% of NTM entering a treatment plant are lost due to coagulation and precipitation of particulates [14], the remaining NTM cells survive disinfection [28] and enter the distribution system where they grow, approximately doubling [28]. Once entering building plumbing, NTM quickly adhere to pipe surfaces, where they grow and form biofilms [29]. As building plumbing has a high ratio of surface to volume NTM numbers are quite high; depending on the type of pipe (e.g., galvanized, copper, stainless steel, or plastic). With an input density of 10,000 colony-forming units/mL of water, upwards of 15,000 colony-forming units/cm^2^ of pipe surface can be attained within 1 h [29]. They also enter water heaters where they adhere to the glass surfaces and find optimal temperatures for growth. Thus, the water heater is a site for the increase in numbers of NTM that are subsequently distributed throughout the structure via pipes. If one calculates the total surface area of the plumbing in a business building, a hospital, home, or even an apartment, one concludes that the surface biofilm of plumbing supports a huge number of microorganisms. Those biofilm NTM populations serve as sources of NTM that “inoculate” the water in the plumbing. Showerhead and tap aerators offer substantial surface areas for adherence and biofilm formation [49]. Further, they are often warmed, and aerated. If the water flow in the building halts and the water stagnates, due to absence of workers, patients, or residents, that does not reduce NTM numbers as they can grow at low oxygen levels [15]. In fact, one of the current concerns associated with closing buildings to prevent COVID-19 transmission is that building plumbing now harbors very high numbers of waterborne opportunistic pathogens. At the very least, building plumbing must be flushed thoroughly.

## 8. A Concluding Challenge

As current water treatment protocols do not substantially reduce NTM numbers, the challenge exists to identify, develop, and implement new disinfection practices. As other opportunistic premise plumbing pathogens—namely *Legionella pneumophila*, *Pseudomonas aeruginosa*, *Acinetobacter baumannii*, and *Stenotrophomonas maltophilia*—share characteristics in common with NTM, any improvement in control of NTM to lower NTM infection would also reduce infections caused by a broad spectrum of opportunistic pathogens.

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
