# Peer review of "Ecology of Nontuberculous Mycobacteria"

_microorganisms, 2021, doi:10.3390/microorganisms9112262_

Round 1

Reviewer 1 Report

A comprehensive and important review of the current state of environmental acquisition of NTM, including the various sources such as soil, dust, water, plumbing, the mechanisms of persistence, and the most common exposures, as well as needed interventions. A few minor comments below:

  • Lines 29-30: The correct citation for the 2014 estimate of 181,00 cases of NTM PD in the US  is:  Strollo et al.  The burden of pulmonary nontuberculous mycobacterial disease in the United States.  Annals 2015 Oct;12(10):1458-64 2014: h (ttps://pubmed.ncbi.nlm.nih.gov/26214350/)
  • Is “sombrero” a technical term? Or does it refer to a hat-like shape?
  • The author discussed the switch from smooth to transparent colonies as a form of adaptation with increased virulence, but what about the transition from smooth to rough? The transition to rough colonies has been identified as a feature of increased virulence.  The authors could consider addressing this issue briefly.
  • Regarding the section “distribution of NTM inferred from skin sensitivity reactions”- the authors describe how their inferences were based on M. scofulaceum, which has since disappeared from the US due to sensitivity to cholorine related to widepspread chlorination of the drinking water system.  My understanding is that PPD-B is the Battey antigen, derived from intracellulare.  Could the author please elaborate on the cross reactivity of M. scofulaceum and M. intracellulare?

Author Response

I am grateful to this Reviewer for their informed and careful review.

  1. I have corrected the citation for number of NTM pulmonary disease cases to Strollo et al. 2015.
  2. I have changed "sombrero" to hat-like shape as suggested.
  3. The manuscript was revised to add "rough" to the description of the transparent-type colonies and added the fact that the rough, transparent are more virulent than the smooth colony-types.
  4. The relevance of the skin sensitivity results have been made clearer by adding the information that PPD-G was prepared from the Gause strain of M. scrofulaceum and there were higher frequencies of PPD-G reactors in the southeastern U.S.

Reviewer 2 Report

In this article entitled “Ecology of nontuberculous mycobactera (NTM)” the author did a thorough review of the current status of knowledge on the environmental infection sources for NTM. Numerous studies have revealed a continuous increase in the worldwide incidence and prevalence of NTM diseases. Better understanding of the niches exploited by NTM and their ecology is essential for preventing NTM infections and developing new methods for their effective treatment and elimination. I do believe this nice review is relevant to the field and should be published after minor revisions. I would also suggest that the author include a graphic presentation, if possible.

Minor comment:

References: please check the order of the references, reference number 37 is cited 2 times.

Author Response

I appreciate the careful review and kind comments from this Reviewer.

1. Thank you for pointing out the mis-numbering of references 37 and 38. Reference 37 (Sax et al., 2015) in the original has remained as number 37. The mis-numbered 37 (Blanchard and Syzdek, 1970) has been re-numbered 38 in the References. The text did not require revision.  

Reviewer 3 Report

The author Joseph O. Falkinham III entitled “Ecology of Nontuberculous Mycobacteria (NTM)” discussed detailed characteristics of NTM which contribute to a deeper understanding of NTM and could help researchers in the development of treatment for NTM infections. Overall this review is well written and includes relevant references. I will recommend the acceptance of this review for the journal microorganisms.

Author Response

I thank this Reviewer for their positive and kind comments.